# The prognostic value of the early neutrophil-to-lymphocyte ratio for 28-day mortality in sepsis patients: A machine learning-based investigation of the MIMIC database

Jiyang Liao[1☋], Qianwen Xiang[2☋], Xingwang Chen[1☋], Long Wu[1], Houwang Chen[1], Zhijun Yao[1], Huachu Wu[1]*, Jianbo Lai[1]*

1 Department of Intensive Care Unit, Shenzhen Hospital of Integrated Traditional Chinese and Western Medicine, Shenzhen, Guangdong Province, China, 2 Department of Cardiovascular Medicine, Shenzhen Hospital of Integrated Traditional Chinese and Western Medicine, Shenzhen, Guangdong Province, China

☋ These authors contributed equally to this study and are co-first authors.
* 574631282@qq.com (HW); szszxyjhyylaijianbo@outlook.com (JL)

## Abstract

### Background

The neutrophil-to-lymphocyte ratio (NLR) has shown inconsistent prognostic value in individuals with sepsis. This study aimed to clarify its ability to predict 28-day mortality via a machine learning-based analysis of a large ICU database.

### Methods

This retrospective analysis employed data from the MIMIC-IV database (v3.1). The Boruta algorithm combined with XGBoost was used for two-stage feature selection. Patients were stratified by NLR quartiles into three groups (low: <4.34, intermediate: 4.34–14.70, and high: >14.70). This study defined 28-day mortality as the primary outcome. Associations between the NLR and mortality were evaluated by using multivariable logistic regression (progressively adjusted for demographic, clinical, and machine learning-derived features), along with restricted cubic splines. Sensitivity analyses included quantifying NLR feature importance via machine learning and performing subgroup analyses across clinical strata.

### Results

This cohort study included 4,376 patients with a 28-day mortality rate of 18.4%. Compared with the SOFA and SAPS II scores, the prediction performance of XGBoost was superior (ROC-AUC 0.875; 95% CI 0.854–0.896; PR-AUC 0.603). Although the NLR ranked 14th in SHAP-based feature importance, multivariable analysis confirmed its independent association with elevated mortality risk: 28-day (OR 1.16; 95% CI 1.06–1.27; p < 0.001), in-hospital (OR 1.13; 95% CI 1.03–1.24; p < 0.001), and

**Data availability statement:** The complete minimal anonymized dataset necessary to replicate all study findings is publicly available from Zenodo (https://doi.org/10.5281/zenodo.20139761).

**Funding:** This project was supported by grants from the High-level Medical Team Project in Baoan (No. 202405) and Medical and Health Scientific Research Project of Shenzhen Baoan District in 2023 (No. 2023JD118). Funding related to No. 2023JD118 was received by JYL, who conceived and designed the study, supervised the research, performed the data collection and curation, writing-review and editing the manuscript. Funding related to 2023. Funding related to No. 202405 was received by JBL, who contributed to critical revision of the manuscript and participated in the drafting of the original manuscript.

**Competing interests:** The authors declare that they have no competing interests.

ICU (OR 1.14; 95% CI 1.03–1.25; p = 0.008). Stratified analyses indicated consistent mortality associations for the NLR, with enhanced predictive value being observed in patients aged >45 years and those with SOFA scores ≤4 or SAPS II scores >29. Specifically, patients ≥65 years of age demonstrated a 17% increase in 28-day mortality risk (p = 0.019), and patients with a SOFA score ≤4 exhibited a greater than 20% elevated risk across all of the endpoints (p < 0.001), whereas no significant association was observed in the SOFA ≥9 subgroup (p = 0.369).

## Conclusions

The NLR effectively identifies inflammation-driven mortality risk for early sepsis patients but fails to predict outcomes for patients with terminal organ failure. This biphasic predictive pattern highlights the unique value of the NLR in moderate sepsis risk stratification but cautions against its use in cases of advanced disease. Its value lies in dynamic monitoring rather than static risk assessment.

## Introduction

Sepsis is a critical medical state involving organ failure resulting from a maladaptive systemic reaction to infection that impacts millions of patients worldwide annually [1]. Despite recent advancements in sepsis recognition and treatment strategies, mortality rates remain alarmingly high and range from 16.7% to 33.3% [2,3], imposing a substantial burden on global healthcare economics. Nevertheless, early identification and intervention have led to significant improvements in clinical outcomes.

Current assessment methods predominantly rely on manual documentation or electronic medical records, which often incorporate nursing process-dependent parameters such as vital sign monitoring [4–7]. Variations in measurement and recording protocols across healthcare institutions may lead to substantial discrepancies in evaluation results. Although the levels of routinely available inflammatory markers such as procalcitonin and C-reactive protein are easily determined in clinical practice, they exhibit limited prognostic value for sepsis because of their suboptimal specificity and sensitivity [1,8]. Although conventional scoring systems such as sequential organ failure assessment (SOFA) and acute physiology and chronic health assessment scoring system II (APACHE II) demonstrate high prognostic value, their complex assessment procedures and poor operational feasibility limit their widespread clinical application [9–11]. Consequently, the pursuit of more objective and convenient prognostic indicators remains important.

Neutrophils (NEUT) and lymphocytes (LYMPH), as the principal cellular components of innate and adaptive immunity, respectively, play indispensable roles as first-line immune defenses during systemic inflammatory responses [12,13]. Bacterial or fungal infections typically induce isolated NEUT elevation with concomitant lymphopenia, whereas viral infections or malignancies may predominantly increase LYMPH counts [14]. Consequently, the neutrophil-to-lymphocyte ratio (NLR) generally increases during inflammatory states. Although clinically favorable because

of its accessibility and low cost, some alterations in the NLR are not inflammation-specific; for example, acute trauma, stroke, myocardial infarction, and postoperative complications can similarly modify this ratio, which compromises its diagnostic specificity and contributes to persistent clinical skepticism regarding its utility [15]. The COVID-19 pandemic has reinvigorated research interest in the NLR [16,17], particularly as a result of a recent meta-analysis by Wu et al. [18], which reported reliable prognostic performance of the NLR in sepsis despite considerable between-study heterogeneity ($I^2 = 87.2\%$, 95% CI 79.5–92; $p < 0.0001$). Moreover, a retrospective study conducted by Schupp et al. [19] failed to demonstrate a discriminative capacity for 30-day mortality. These contradictory prognostic interpretations in sepsis underscore the need for machine learning (ML)-based reevaluations of the ability of the NLR to predict mortality.

Machine learning algorithms have emerged in recent years as powerful tools in clinical prediction models. The 2021 sepsis guidelines highlight the superior discriminative capacity of ML algorithms for predicting in-hospital mortality in sepsis patients [1]. These algorithms offer dual advantages—the efficient management of missing data and the ability to integrate weak predictive features into robust models through data-driven learning. Among the various ML techniques, extreme gradient boosting (XGBoost) has demonstrated exceptional data mining and predictive modeling capabilities. Its dominance is evidenced by its adoption in 17 of 29 winning solutions in the 2015 Kaggle competitions and its exclusive use by all top 10 teams that participated in the 2015 KDD Cup. Several studies have validated the clinical utility of XGBoost: Dong et al. developed a ML model that predicts pediatric acute kidney injury 48 hours in advance by analyzing premorbid physiological measurements and incorporating current guidelines; Zhang et al. established an XGBoost model that effectively distinguishes between fluid-responsive and nonresponsive sepsis patients; and Yue et al. created a ML model for the early detection of sepsis-associated acute kidney injury and identified XGBoost as the optimal predictive algorithm. These findings collectively support the theoretical potential of ML algorithms to enhance predictive model development and validation in critical care. However, no studies have yet employed XGBoost modeling with large datasets to systematically evaluate the ability of the NLR to predict 28-day mortality, which could provide valuable insights into its clinical relevance under resource-constrained conditions.

## Methods

### Database

The patient data utilized for this study were derived from the Medical Information Mart for Intensive Care IV database version 3.1 (MIMIC-IV v3.1) [20,21]. This dataset, which includes deidentified health data from more than 65,000 Intensive Care Unit (ICU) hospitalizations and over 200,000 emergency department visits at Beth Israel Deaconess Medical Center (Boston, MA) from 2008–2022, offers broad clinical coverage. The dataset includes demographic characteristics, physiological parameters validated by ICU nursing staff, laboratory test results, therapeutic intervention orders, standardized data dictionaries, and diagnostic codes (International Classification of Diseases, 9th and 10th Revisions; ICD-9 and ICD-10). As this study used a deidentified public database with all protected health information removed, this study was exempted from the requirement to obtain individual patient informed consent from the Institutional Review Board (IRB) of the Massachusetts Institute of Technology (MIT). Author JY Liao (certification ID: 63692240) completed the required training and was authorized to access the database for research purposes. This study was performed in accordance with the Declaration of Helsinki. Due to the deidentified nature and public availability of the MIMIC database (which contains no personally sensitive information), the study was ethically exempt from review.

### Study population

Inclusion criteria for this study were as follows: (1) aged between 18 and 85 years; (2) an ICU stay duration that exceeded 24 hours; and (3) suspected or confirmed infection with a SOFA score ≥2. For patients with multiple ICU admissions, only data obtained from the first ICU stay were included in the analysis. Patients were excluded if they met any of the following

conditions: (1) had missing laboratory data, including NEUT, LYMPH and platelet (PLT) data; or (2) had a documented diagnosis of malignant tumors or autoimmune diseases. The study design is schematically illustrated in Fig 1.

## Data extraction

Using structured query language (SQL) within the Navicat Premium environment (version 16.1.12), we extracted the requisite patient data from the MIMIC-IV database. The analysis focused exclusively on each patient's first ICU admission record. Due to the extensive temporal data entries in the MIMIC-IV database, strict time windows were applied during data extraction to minimize potential confounding from different treatment phases. Demographic characteristics and clinical variables, such as sex, age, weight, SOFA score, simplified acute physiology score II (SAPS II) score and comorbidities, were extracted from the earliest available clinical records during the first 24 hours following ICU admission. Vital signs were collected within the first 2 hours after ICU admission. The crystalloid fluid intake volume was obtained within the first 3 hours and was used to calculate the crystalloid-to-weight ratio. The first available laboratory test results obtained within the initial 6 hours of ICU admission were extracted for analysis. Cases with NEUT, LYMPH, or PLT counts of zero were excluded to ensure valid calculation of the systemic immune-inflammation index (SII), which was derived using the following formula: absolute NEUT count × PLT count/absolute LYMPH count [22]. Fluid intake and output volumes were recorded during the first 24 hours of ICU admission, after which the fluid intake-to-weight ratio was calculated. The

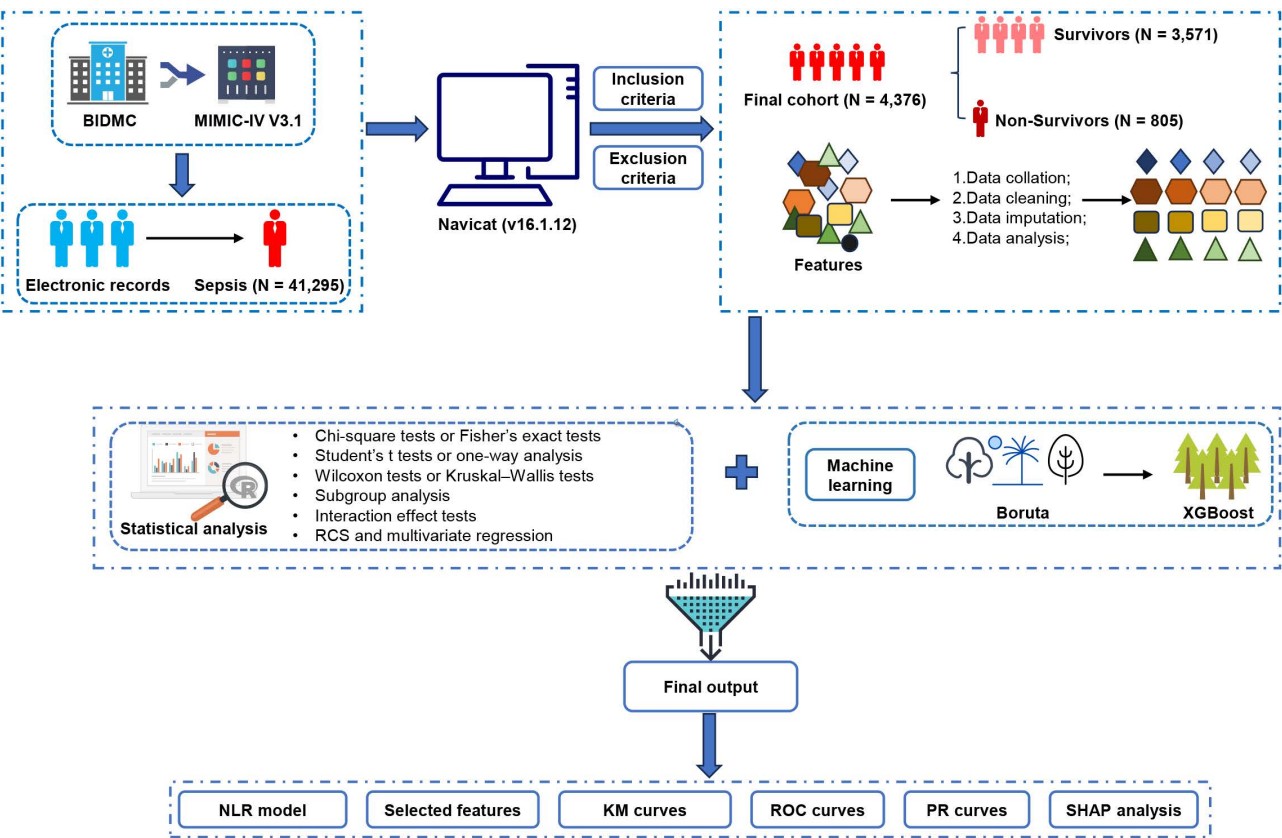

**Fig 1. Overview of the study design and workflow.** BIDMC, Beth Israel Deaconess Medical Center; MIMIC-IV, Medical Information Mart for Intensive Care IV; XGBoost, extreme gradient boosting; NLR, neutrophil-to-lymphocyte ratio; RCS, restricted cubic splines; ROC, receiver operating characteristic; PR, precision-recall; SHAP, Shapley additive explanations.

vasoactive inotropic score (VIS) and antibiotic initiation time were concurrently determined. Upon ICU discharge, the types of mechanical ventilation (MV) and continuous renal replacement treatment (CRRT) were recorded. Patients who received noninvasive ventilation, invasive ventilation, or tracheostomy were classified as mechanically ventilated. The primary outcome was defined as all-cause mortality within 28 days following ICU admission. Secondary outcomes included in-hospital mortality, ICU mortality, length of hospital stay, and length of ICU stay. Follow-up was conducted from ICU admission until 28 days thereafter.

To address missing data, a 40% missingness threshold was set. Consequently, any parameter with missing values exceeding this percentage was excluded from analysis under the assumption that the data were missing at random (MAR). [23]. For variables where fewer than 40% of the values were missing, we performed imputation using the "miss-Forest" package (version 1.5) in R Studio [24]. To assess the robustness of the imputation, we compared the distribution of key variables before and after imputation by using the missForest algorithm. Post-imputation values remained within clinically plausible ranges, and no substantial shifts in central tendency or dispersion were observed (S1 and S2 Tables).

### Feature selection

Before the association between the NLR and 28-day mortality was assessed, we implemented a rigorous two-stage ML-driven feature selection protocol. In the first phase, we performed 10 iterative runs of the Boruta algorithm to evaluate feature selection stability. During each run, features were classified as "confirmed" if their importance significantly exceeded the maximum shadow feature importance ($p < 0.05$) [25]. Feature stability was quantified by calculating the selection frequency (SF), which is defined as the ratio of confirmed designations to total runs. Only features that demonstrated high stability (SF ≥ 0.9) were retained for subsequent XGBoost analysis [26]. The XGBoost feature selection phase employed Pearson correlation-based collinearity analysis to remove redundant features, followed by hyperparameter optimization via grid search with three repetitions of 5-fold cross-validation, learning curve analysis, and early stopping to prevent overfitting—all of which were implemented to enhance clinical interpretability and enable SHAP analysis. To address class imbalance in sepsis mortality outcomes, the precision-recall area under the curve (PR-AUC) was employed as a more informative metric compared to the conventional receiver operating characteristic AUC (ROC-AUC) for evaluating model performance, with particular emphasis being focused on the accurate detection of the minority class. The model's performance was then comprehensively assessed by using a suite of metrics, including accuracy, sensitivity, specificity, and F1-score metrics, in order to ensure a robust evaluation of its clinical utility. The final model was interpreted using Shapley additive explanations (SHAP) to assess the magnitude and direction of the individual feature influences [27,28]; additionally, the top-ranking NLR-associated variables were subsequently entered into multivariable regression analysis for clinical validation.

### Statistical analysis

Statistical analyses and visualizations were performed by using the R programming language (version 4.4.3), with the exception of XGBoost modeling, which was implemented in Python (version 3.12). A two-sided p value <0.05 was considered to be statistically significant. Categorical variables were analyzed by using chi-square tests or Fisher's exact tests and are presented as frequencies (percentages). To control the false discovery rate (FDR) due to multiple comparisons across categorical covariates (e.g., antibiotic initiation, dialysis type and ventilation status), p values were adjusted by using the Benjamini–Hochberg procedure. For continuous variables, we assessed normality by histograms and skewness analysis (absolute skewness ≥2 with evident left or right skewing in histograms was considered to indicate a nonnormal distribution). Normally distributed continuous variables were compared using Student's t tests or one-way analysis of variance, whereas nonnormally distributed variables were assessed via Wilcoxon or Kruskal–Wallis tests, with the results being expressed as the means ± standard deviations (SDs) or medians (interquartile ranges, or IQRs). We examined potential nonlinear associations between the NLR and 28-day mortality using restricted cubic splines (RCS) regression,

which tests the statistical significance of nonlinear terms. We simultaneously generated PR and ROC curves for the NLR, XGBoost model, and both the SOFA and SAPS II scoring systems and calculated the Youden index to determine the optimal cutoff values and evaluate their ability to predict 28-day mortality. Multivariate regression models were used to assess the associations between the NLR and 28-day mortality, hospital mortality, and ICU mortality. All included continuous variables were standardized using Z scores to minimize scale differences. Three sequential models were constructed: Model 1 included only the standardized NLR, Model 2 added sex, standardized age, heart rate (HR), respiratory rate (RR), weight, and mean arterial pressure (MAP) to Model 1, and Model 3 further incorporated clinically relevant variables, previously established predictors, and features selected through the Boruta algorithm and XGBoost modeling. Subgroup analyses were performed to further examine the NLR–mortality relationship, with subgroups defined according to clinical relevance and data distribution patterns. Interaction terms were included to assess subgroup heterogeneity, and Firth's penalized likelihood regression was applied to prevent false-positive results in small sample sizes. Interaction p values were also adjusted for the FDR to comprehensively evaluate potential effect modifiers and the robustness of the NLR predictive value. Survival was analyzed by using Kaplan–Meier curves with log-rank tests for between-group comparisons and Cox proportional hazards models, with the results being reported as hazard ratios with 95% confidence intervals.

## Results

### Baseline characteristics

In this study, 4,376 eligible patients from the MIMIC-IV database (with 28-day, in-hospital, and ICU mortality rates of 18.40%, 17.50%, and 14.12%, respectively) were ultimately enrolled. The demographic characteristics, vital signs, laboratory tests, organ dysfunction assessments, treatments and other outcomes are summarized in Table 1. Compared with nonsurvivors, survivors were significantly younger ($59.77 \pm 14.97$ vs. $63.37 \pm 15.10$; $p < 0.05$), were more likely to be male (63.93% vs. 59.25%), and had lower disease severity, as reflected by SOFA ($3.76 \pm 1.96$ vs. $4.99 \pm 2.71$) and SAPS II scores ($36.79 \pm 12.90$ vs. $51.74 \pm 15.76$). With respect to vital signs, survivors demonstrated lower HR ($89.48 \pm 17.68$ vs. $95.55 \pm 20.99$ bpm; $p < 0.001$), lower RR ($19.07 \pm 5.54$ vs. $22.11 \pm 6.29$ breaths/min; $p < 0.001$), and higher percutaneous oxygen saturation ($SpO_2$, median: 98.57% vs. 97.00%; $p < 0.001$) than nonsurvivors, whereas MAP did not significantly differ between the groups ($81.58 \pm 15.22$ vs. $81.81 \pm 19.81$ mmHg; $p = 0.755$). Although survivors demonstrated marginally better vital signs (all $p < 0.05$ except $SpO_2$), the intergroup differences were clinically modest. Laboratory analyses revealed significantly elevated inflammatory markers, including the NLR (6.97 [IQR 4.11–12.59] vs. 12.70 [IQR 6.92–22.75]; $p < 0.05$) and the SII, alongside more pronounced acidotic patterns on blood gas analysis, in nonsurvivors. Higher creatinine and blood urea nitrogen (BUN) levels suggested increased renal dysfunction in nonsurvivors, which is consistent with the results of the organ dysfunction assessments. Erythrocyte and hemoglobin-related parameters significantly differed between the two groups. Organ dysfunction distributions revealed greater respiratory and coagulation dysfunction in survivors, whereas nonsurvivors had greater hepatic, neurological, and renal failure rates (all $p < 0.05$). Comorbidity analyses revealed significant differences in hypertension, heart failure, chronic obstructive pulmonary disease, chronic kidney disease, atrial fibrillation, cerebral infarction, cerebral hemorrhage, and thrombosis. Fluid resuscitation metrics indicated minimal crystalloid administration within the first 3 ICU hours in both groups, with survivors receiving less crystalloid. Although statistically significant differences were observed in weight-adjusted crystalloid ratios, these differences lacked clinical relevance. Analysis of 24-hour fluid balance revealed higher total intake and output volumes in survivors than in nonsurvivors, whereas nonsurvivors exhibited greater net positive fluid balance, suggesting potential fluid overload in this group. Nonsurvivors required more MV, CRRT, and vasopressor support (all $p < 0.05$), whereas survivors had higher rates of antibiotic initiation within 1 hour of ICU admission. Although survivors had shorter ICU stays, their total hospitalization duration exceeded that of nonsurvivors.

On the basis of the overall NLR distribution and quartile values, patients were categorized into low (<4.34), intermediate (4.34–14.70), and high (>14.70) NLR groups (as shown in Table 2). All mortality outcomes, including 28-day, hospital,

**Table 1. Baseline characteristics of the survivor and nonsurvivor groups.**

| Categories | Survivors (N = 3,571) | Non-survivors (N = 805) | P-value |
|---|---|---|---|
| **Demographic** | | | |
| Age, years | 59.77 ± 14.97 | 63.37 ± 15.10 | <0.001 |
| Gender(male), n (%) | 2283 (63.93%) | 477 (59.25%) | 0.012 |
| SOFA | 3.76 ± 1.96 | 4.99 ± 2.71 | <0.001 |
| SAPS II | 36.79 ± 12.90 | 51.74 ± 15.76 | <0.001 |
| Weight, kg | 89.12 ± 23.70 | 87.44 ± 27.64 | 0.111 |
| **Vital Signs** | | | |
| Heart Rate, bpm | 89.48 ± 17.68 | 95.55 ± 20.99 | <0.001 |
| Respiratory Rate, bpm | 19.07 ± 5.54 | 22.11 ± 6.29 | <0.001 |
| MAP, mmHg | 81.58 ± 15.22 | 81.81 ± 19.81 | 0.755 |
| SpO2, % | 98.57 (96.00–99.40) | 97.00 (94.00–100.00) | <0.001 |
| **Laboratory Tests** | | | |
| Platelets, K/uL | 185.71 ± 102.27 | 192.85 ± 115.18 | 0.106 |
| Neutrophils, K/uL | 10.84 ± 6.05 | 13.53 ± 8.30 | <0.001 |
| Lymphocytes, K/uL | 1.34 (0.81–2.01) | 0.91 (0.52–1.50) | <0.001 |
| WBC, K/uL | 13.34 ± 6.72 | 16.08 ± 9.60 | <0.001 |
| NLR | 6.97 (4.11–12.59) | 12.70 (6.92–22.75) | <0.001 |
| SII | 1113.95 (579.22–2470.93) | 2083.27 (913.42–4536.00) | <0.001 |
| PO2, mmHg | 221.07 ± 115.04 | 151.38 ± 86.71 | <0.001 |
| PCO2, mmHg | 41.00 (37.80–45.00) | 40.92 (36.30–46.44) | 0.716 |
| pH | 7.36 ± 0.08 | 7.31 ± 0.12 | <0.001 |
| BE, mEq/L | −1.22 ± 4.19 | −4.22 ± 6.29 | <0.001 |
| Bicarbonate, mEq/L | 22.24 ± 4.25 | 20.40 ± 5.88 | <0.001 |
| BUN, mg/dL | 17.00 (12.00–26.00) | 31.00 (19.00–52.00) | <0.001 |
| Creatinine, mg/dL | 0.90 (0.70–1.30) | 1.60 (1.00–2.70) | <0.001 |
| Hemoglobin, g/dL | 10.51 ± 2.29 | 10.91 ± 2.68 | <0.001 |
| Hematocrit, % | 31.75 ± 6.83 | 33.71 ± 8.12 | <0.001 |
| RBC, m/uL | 3.50 ± 0.78 | 3.61 ± 0.93 | 0.001 |
| RDW, % | 14.34 ± 1.91 | 15.91 ± 2.85 | <0.001 |
| MCH, pg | 30.18 ± 2.53 | 30.47 ± 3.04 | 0.013 |
| MCHC, g/dL | 33.12 ± 1.59 | 32.38 ± 1.82 | <0.001 |
| **Organ Dysfunction, n (%)** | | | |
| Respiration | 1564 (43.8%) | 250 (31.06%) | <0.001 |
| Coagulation | 1528 (42.79%) | 285 (35.4%) | <0.001 |
| Hepatic | 494 (13.83%) | 234 (29.07%) | <0.001 |
| Cardiovascular | 2289 (64.1%) | 507 (62.98%) | 0.559 |
| Neurologic | 543 (15.21%) | 145 (18.01%) | 0.048 |
| Kidney | 1051 (29.43%) | 435 (54.04%) | <0.001 |
| **Comorbidities, n (%)** | | | |
| Diabetes | 982 (27.5%) | 247 (30.68%) | 0.082 |
| Hypertension | 1786 (50.01%) | 361 (44.84%) | 0.007 |
| Heart Failure | 845 (23.66%) | 252 (31.3%) | <0.001 |
| AMI | 49 (1.37%) | 18 (2.24%) | 0.086 |
| COPD | 171 (4.79%) | 74 (9.19%) | <0.001 |
| CKD | 459 (12.85%) | 192 (23.85%) | <0.001 |
| Atrial Fibrillation | 979 (27.42%) | 270 (33.54%) | 0.001 |

*(Continued)*

**Table 1.** (Continued)

| Categories | Survivors (N = 3,571) | Non-survivors (N = 805) | P-value |
|---|---|---|---|
| Cerebral Infarction | 335 (9.38%) | 125 (15.53%) | <0.001 |
| Cerebral Hemorrhage | 45 (1.26%) | 36 (4.47%) | <0.001 |
| Thrombosis | 372 (10.42%) | 138 (17.14%) | <0.001 |
| **Treatment** | | | |
| Crystalloid Volume in 3 h, mL | 80.50 (0.00–350.00) | 153.00 (0.00–558.40) | <0.001 |
| Fluid/Weight in 3 h, mL/kg | 0.85 (0.00–4.24) | 1.89 (0.00–6.74) | <0.001 |
| Fluid Input in 24 h, mL | 4939.43 (2247.60–7224.11) | 3224.80 (1327.82–6880.24) | <0.001 |
| Fluid Output in 24 h, mL | 2580.00 (1755.00–3540.00) | 1370.00 (610.00–2520.00) | <0.001 |
| Fluid Balance in 24 h, mL | 2373.99 ± 4304.62 | 3022.37 ± 5097.93 | <0.001 |
| Fluid Input/Weight in 24 h, mL/kg | 57.01 (24.69–86.47) | 38.11 (16.38–79.78) | <0.001 |
| Antibiotic Initiation (less than 1 h), n (%) | 1074 (30.1%) | 172 (21.4%) | <0.001 |
| VIS | 0.00 (0.00–5.00) | 8.60 (0.00–40.06) | <0.001 |
| CRRT, n (%) | 174 (4.9%) | 234 (29.1%) | <0.001 |
| Mechanical Ventilation, n (%) | 2422 (67.8%) | 604 (75.0%) | <0.001 |
| **Outcomes** | | | |
| LOS of Hospital, days | 8.84 (5.66–15.75) | 6.71 (2.93–12.98) | <0.001 |
| LOS of ICU, days | 2.94 (1.62–6.20) | 4.93 (2.48–9.29) | <0.001 |

Data: Mean ± Standard Deviation or Median (Q1–Q3) or N (%). P-value < 0.05 was considered statistical significance.

SOFA, Sequential Organ Failure Assessment; SAPS II, Simplified Acute Physiology Score II; MAP, Mean Arterial Pressure; $SpO_2$, Peripheral Oxygen Saturation; WBC, White Blood Cell; NLR, Neutrophil-to-Lymphocyte Ratio; SII, Systemic Immune-Inflammation Index; $PO_2$, Partial Pressure of Oxygen; $PCO_2$, Partial Pressure of Carbon Dioxide; BE, Base Excess; BUN, Blood Urea Nitrogen; RBC, Red Blood Cell; RDW, Red Cell Distribution Width; MCH, Mean Corpuscular Hemoglobin; MCHC, Mean Corpuscular Hemoglobin Concentration; AMI, Acute Myocardial Infarction; COPD, Chronic Obstructive Pulmonary Disease; CKD, Chronic Kidney Disease; CRRT, Continuous Renal Replacement Therapy; VIS, Vasoactive-Inotropic Score; LOS, Length of Stay.

and ICU mortality, significantly increased in a stepwise manner with increasing NLR (p < 0.05) until they reached approximately 30% in the high-NLR group. Similarly, both hospital and ICU length of stay increased in a concentration-dependent manner (p < 0.05). No statistically significant differences were observed in mean corpuscular hemoglobin (MCH) levels, cardiovascular failure incidence, or the incidence of comorbidities such as diabetes, atrial fibrillation, or cerebral infarction. However, all other parameters that demonstrated significant differences between the survivor and nonsurvivor groups maintained concentration-dependent statistical significance across NLR stratifications (p < 0.05).

### Selection of features in the models

The results from a single Boruta run are presented in S1 Fig, whereas Fig 2A displays stability-selected features (SF ≥ 0.9) from 10 iterations; this process identified 37 confirmed and 2 tentative features, whereas the remaining features were rejected. These 39 features were subjected to XGBoost modeling, which was preceded by collinearity analysis (Pearson's r ≥ 0.7 exclusion) and clinical relevance screening (S2 Fig and S3 Table), thus yielding 29 final parameters. As shown in S4 Table, the optimized XGBoost model achieved a test-set ROC-AUC of 0.875 (95% CI: 0.854–0.896) and a PR-AUC of 0.603 (95% CI: 0.534–0.671), thereby outperforming the individual NLR, as well as the SAPS II and SOFA scores, for 28-day mortality prediction (Fig 3A, 3B). The data in Fig 2B reveal 13 important parameters, and Model 3 ultimately incorporated SAPS II and SOFA scores, as well as 24-hour fluid output, VIS, 24-hour fluid balance, dialysis type, cerebral infarction, atrial fibrillation, red cell distribution width (RDW), partial pressure of oxygen ($PO_2$), hematocrit, creatinine, LYMPH, base excess (BE), and MCH. S5 Table presents the predictive contribution ranking of the NLR in the XGBoost

**Table 2. Characteristics and outcomes of patients categorized by NLR.**

| Categories | NLR < 4.34 (N = 1,094) | NLR 4.34–14.70 (N = 2185) | NLR > 4.34 (N = 1,094) | P-value |
|---|---|---|---|---|
| **Demographic** | | | | |
| Age, years | 61.20 ± 14.44 | 59.87 ± 15.21 | 60.79 ± 15.32 | 0.039 |
| Gender (male), *n* (%) | 665 (60.79%) | 1445 (66.04%) | 650 (59.41%) | <0.001 |
| SOFA | 3.76 ± 1.96 | 3.92 ± 2.10 | 4.36 ± 2.44 | <0.001 |
| SAPS II | 36.60 ± 13.33 | 38.63 ± 14.17 | 44.30 ± 15.76 | <0.001 |
| Weight, kg | 86.73 ± 22.50 | 89.91 ± 23.44 | 88.70 ± 28.02 | <0.001 |
| **Vital Signs** | | | | |
| Heart Rate, bpm | 85.27 ± 16.05 | 90.75 ± 17.83 | 95.61 ± 20.49 | <0.001 |
| Respiratory Rate, bpm | 17.44 ± 4.52 | 19.46 ± 5.69 | 22.16 ± 6.20 | <0.001 |
| MAP, mmHg | 80.29 ± 11.95 | 81.83 ± 16.16 | 82.54 ± 19.44 | <0.001 |
| SpO2, % | 98.92 (97.66–99.47) | 98.52 (96.00–99.75) | 97.00 (94.00–99.00) | <0.001 |
| **Laboratory Tests** | | | | |
| Platelets, K/uL | 159.18 ± 81.02 | 189.82 ± 102.86 | 209.29 ± 122.19 | <0.001 |
| Neutrophils, K/uL | 6.84 ± 3.81 | 11.12 ± 5.22 | 16.25 ± 7.83 | <0.001 |
| Lymphocytes, K/uL | 2.09 (1.48–2.86) | 1.34 (0.96–1.84) | 0.57 (0.37–0.84) | <0.001 |
| WBC, K/uL | 9.93 ± 5.30 | 13.63 ± 6.38 | 18.20 ± 8.71 | <0.001 |
| SII | 440.61 (291.56–620.37) | 1267.47 (798.13–2038.41) | 4795.85 (2935.87–7612.25) | <0.001 |
| PO2, mmHg | 261.23 ± 119.21 | 209.82 ± 111.95 | 152.14 ± 80.15 | <0.001 |
| PCO2, mmHg | 41.34 ± 7.96 | 42.10 ± 8.69 | 42.82 ± 11.27 | 0.001 |
| pH | 7.38 ± 0.09 | 7.36 ± 0.09 | 7.33 ± 0.09 | <0.001 |
| BE, mEq/L | 0.00 (−1.84–1.71) | −0.68 (−3.67–1.00) | −3.00 (−6.00–−0.35) | <0.001 |
| Bicarbonate, mEq/L | 22.43 ± 3.86 | 22.15 ± 4.42 | 20.90 ± 5.56 | <0.001 |
| BUN, mg/dL | 16.00 (12.00–21.00) | 18.00 (13.00–28.36) | 25.50 (16.00–47.00) | <0.001 |
| Creatinine, mg/dL | 0.85 (0.70–1.10) | 1.00 (0.70–1.50) | 1.30 (0.90–2.40) | <0.001 |
| Hemoglobin, g/dL | 9.98 ± 2.06 | 10.63 ± 2.37 | 11.11 ± 2.53 | <0.001 |
| Hematocrit, % | 30.11 ± 6.15 | 32.15 ± 7.04 | 34.05 ± 7.64 | <0.001 |
| RBC, m/uL | 3.31 ± 0.71 | 3.53 ± 0.81 | 3.71 ± 0.87 | <0.001 |
| RDW, % | 13.70 (13.00–14.80) | 14.00 (13.20–15.30) | 14.60 (13.50–16.10) | <0.001 |
| MCH, pg | 30.32 ± 2.39 | 30.24 ± 2.62 | 30.12 ± 2.87 | 0.225 |
| MCHC, g/dL | 33.14 ± 1.55 | 33.09 ± 1.67 | 32.63 ± 1.70 | <0.001 |
| **Organ Dysfunction, *n* (%)** | | | | |
| Respiration | 512 (46.8%) | 967 (44.2%) | 335 (30.62%) | <0.001 |
| Coagulation | 611 (55.85%) | 875 (39.99%) | 327 (29.89%) | <0.001 |
| Hepatic | 121 (11.06%) | 340 (15.54%) | 267 (24.41%) | <0.001 |
| Cardiovascular | 720 (65.81%) | 1387 (63.39%) | 689 (62.98%) | 0.304 |
| Neurologic | 147 (13.44%) | 349 (15.95%) | 192 (17.55%) | 0.028 |
| Kidney | 244 (22.3%) | 717 (32.77%) | 525 (47.99%) | <0.001 |
| **Comorbidities, *n* (%)** | | | | |
| Diabetes | 302 (27.61%) | 610 (27.88%) | 317 (28.98%) | 0.741 |
| Hypertension | 590 (53.93%) | 1080 (49.36%) | 477 (43.6%) | <0.001 |
| Heart Failure | 228 (20.84%) | 532 (24.31%) | 337 (30.8%) | <0.001 |
| AMI | 6 (0.55%) | 39 (1.78%) | 22 (2.01%) | 0.008 |
| COPD | 45 (4.11%) | 105 (4.8%) | 95 (8.68%) | <0.001 |
| CKD | 130 (11.88%) | 302 (13.8%) | 219 (20.02%) | <0.001 |
| Atrial Fibrillation | 297 (27.15%) | 630 (28.79%) | 322 (29.43%) | 0.464 |

*(Continued)*

**Table 2.** (Continued)

| Categories | NLR < 4.34 (N = 1,094) | NLR 4.34–14.70 (N = 2185) | NLR > 4.34 (N = 1,094) | P-value |
|---|---|---|---|---|
| Cerebral Infarction | 113 (10.33%) | 222 (10.15%) | 125 (11.43%) | 0.516 |
| Cerebral Hemorrhage | 11 (1.01%) | 39 (1.78%) | 31 (2.83%) | 0.006 |
| Thrombosis | 78 (7.13%) | 237 (10.83%) | 195 (17.82%) | <0.001 |
| **Treatment** | | | | |
| Crystalloid Volume in 3 h, mL | 50.00 (0.00–320.25) | 98.45 (0.00–393.51) | 106.38 (0.00–508.32) | <0.001 |
| Fluid/Weight in 3 h, mL/kg | 0.64 (0.00–3.86) | 0.98 (0.00–4.60) | 1.35 (0.00–6.15) | <0.001 |
| Fluid Input in 24 h, mL | 5306.60 (2871.38–7256.89) | 4886.59 (2117.32–7349.11) | 3394.81 (1347.47–6590.64) | <0.001 |
| Fluid Output in 24 h, mL | 2700.00 (1960.50–3563.25) | 2505.00 (1590.00–3483.50) | 1807.50 (1015.00–2930.00) | <0.001 |
| Fluid Balance in 24 h, mL | 2510.63 (347.47–4356.38) | 2260.98 (−101.81–4497.61) | 1325.28 (−520.00–4584.98) | <0.001 |
| Fluid Input/Weight in 24 h, mL/kg | 61.92 (33.08–88.92) | 55.86 (22.91–86.79) | 39.45 (15.66–76.70) | <0.001 |
| Antibiotic Initiation (less than 1 h), n (%) | 388 (35.47%) | 625 (28.56%) | 233 (21.3%) | <0.001 |
| VIS | 0.00 (0.00–3.00) | 0.00 (0.00–8.00) | 0.00 (0.00–20.00) | <0.001 |
| CRRT, n (%) | 54 (4.94%) | 176 (8.04%) | 178 (16.27%) | <0.001 |
| Mechanical Ventilation, n (%) | 783 (71.57%) | 1535 (70.16%) | 708 (64.72%) | 0.001 |
| **Outcomes** | | | | |
| LOS of Hospital, days | 7.10 (5.08–11.49) | 8.47 (5.23–14.69) | 10.84 (5.89–19.71) | <0.001 |
| LOS of ICU, days | 2.32 (1.38–4.44) | 3.18 (1.81–6.85) | 4.75 (2.43–9.77) | <0.001 |
| Hospital Mortality, n (%) | 93 (8.5%) | 333 (15.22%) | 340 (31.08%) | <0.001 |
| ICU Mortality, n (%) | 79 (7.22%) | 267 (12.2%) | 272 (24.86%) | <0.001 |
| 28-day Mortality, n (%) | 101 (9.23%) | 347 (15.86%) | 357 (32.63%) | <0.001 |

Data: Mean ± Standard Deviation or Median (Q1–Q3) or N (%). P-value < 0.05 were considered statistical significance.

SOFA, Sequential Organ Failure Assessment; SAPS II, Simplified Acute Physiology Score II; MAP, Mean Arterial Pressure; SpO$_2$, Peripheral Oxygen Saturation; WBC, White Blood Cell; NLR, Neutrophil-to-Lymphocyte Ratio; SII, Systemic Immune-Inflammation Index; PO$_2$, Partial Pressure of Oxygen; PCO$_2$, Partial Pressure of Carbon Dioxide; BE, Base Excess; BUN, Blood Urea Nitrogen; RBC, Red Blood Cell; RDW, Red Cell Distribution Width; MCH, Mean Corpuscular Hemoglobin; MCHC, Mean Corpuscular Hemoglobin Concentration; AMI, Acute Myocardial Infarction; COPD, Chronic Obstructive Pulmonary Disease; CKD, Chronic Kidney Disease; CRRT, Continuous Renal Replacement Therapy; VIS, Vasoactive-Inotropic Score; LOS, Length of Stay; ICU, Intensive Care Unit.

model for 28-day mortality prediction. The NLR ranked 8th in global importance but decreased to 14th in terms of the SHAP-based marginal contribution, which indicates its relatively low individual impact and potential synergistic interactions with other parameters.

## NLR and 28-day, hospital, and ICU mortality

Multivariable logistic regression analysis demonstrated that the standardized NLR was independently associated with 28-day mortality (OR 1.16; 95% CI 1.06–1.27; p < 0.001), in-hospital mortality (OR 1.13; 95% CI 1.03–1.24; p < 0.001), and ICU mortality (OR 1.14; 95% CI 1.03–1.25; p = 0.008) after full adjustment (Model 3, Table 3). Unadjusted analysis by NLR quartiles revealed a dose–response relationship with all of the mortality endpoints (S3A–C Fig), and stronger associations were observed with increasing NLR values. In the fully adjusted model (Model 3), patients in the highest NLR quartile exhibited significantly higher risks of 28-day mortality (OR 1.95, 95% CI 1.40–2.75), in-hospital mortality (OR 1.90, 95% CI 1.34–2.70), and ICU mortality (OR 1.79, 95% CI 1.23–2.62) compared to those in the lowest quartile, and all of the trend tests were statistically significant (p < 0.001).(Fig 3C) temporally corroborated these findings, which demonstrates early divergence among groups by day 7 (log-rank p < 0.0001), whereas the high-NLR group exhibited markedly decreased survival. Restricted cubic spline analysis (S4 Fig) demonstrated biphasic nonlinear relationships between the NLR and

 

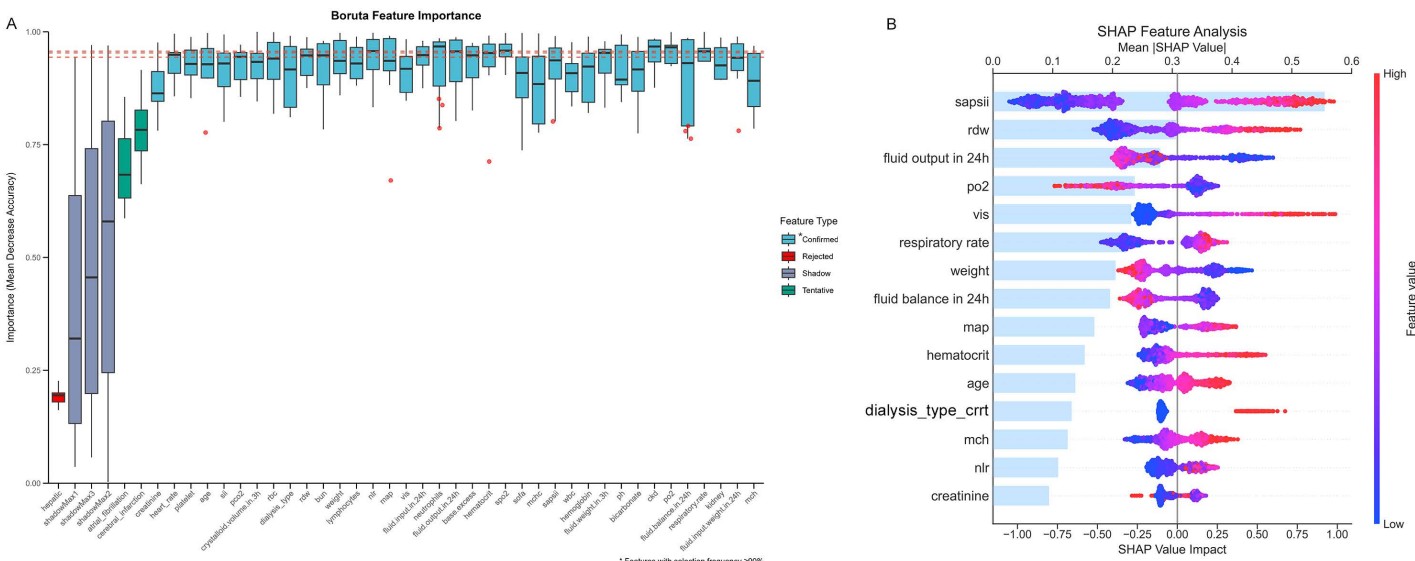

**Fig 2. Application of machine learning algorithm for feature selection. A** Feature stability selection for the relationship between the NLR and 28-day mortality was analyzed by using the Boruta algorithm. The x-axis lists parameters (actual and shadow features), whereas the y-axis shows noise-adjusted importance scores from 10 Boruta runs. Features are color-coded by stability: light blue (confirmed, SF ≥ 90%), green (tentative, 50% ≤ SF < 90%), red (rejected, SF < 50%), and gray (shadow features). A red dashed line marks the stability threshold. **B** SHAP analysis of XGBoost model predictions. Features are vertically ranked by the mean absolute SHAP value (light blue bars), with individual SHAP values (colored dots) distributed horizontally and colored by normalized feature magnitude (blue: low, red: high). The gray dashed line marks the baseline.

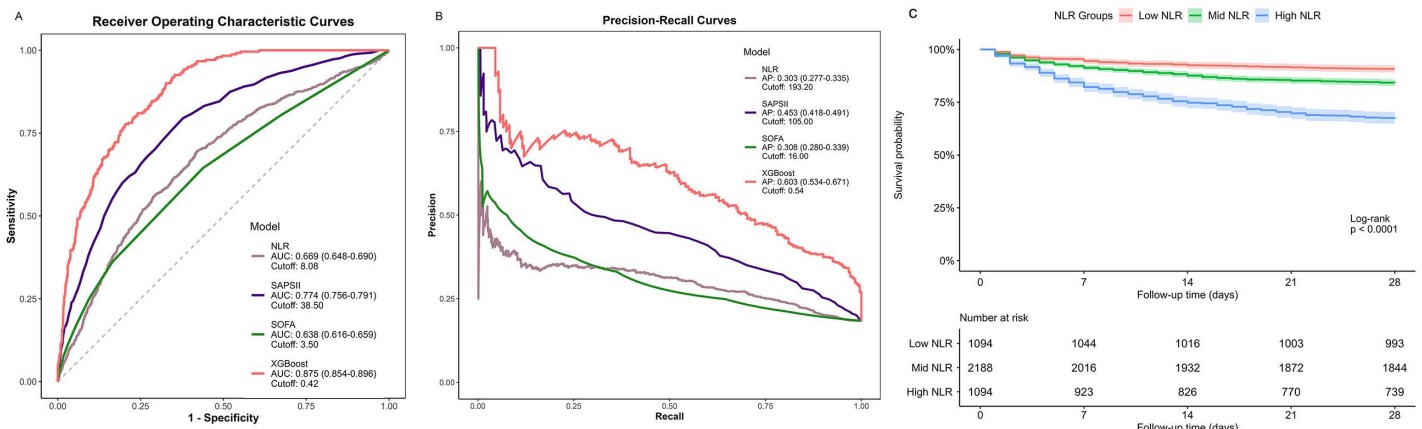

**Fig 3. Comparative analysis of predictive models and survival stratification by the NLR on 28-day mortality.** Panel A: receiver operating characteristic (ROC) curves demonstrate the discriminative performance of the NLR, SAPS II, SOFA, and XGBoost; Panel B: precision-recall (PR) curves assessing the trade-off between precision and recall; Panel C: Kaplan-Meier survival analysis stratified by NLR quartiles.

all of the mortality endpoints, with Wald tests confirming significant nonlinear effects across all of the models (P-nonlinear <0.05). The ROC-derived threshold (NLR = 7.3) coincided with both the median NLR value and the first inflection point, where the predicted probabilities sharply increased. This predictive probability plateaued near the second inflection point (NLR = 27.3) and increased marginally from 0.15 to 0.30, even though sample representation markedly declined in this range. Beyond the second inflection point, statistical reliability diminished because of sparse data.

**Table 3. The association between NLR groups and 28-day mortality, hospital mortality and ICU mortality.**

| Exposure | Model 1 | | Model 2 | | Model 3 | |
|---|---|---|---|---|---|---|
| | OR (95% CI) | P-value | OR (95% CI) | P-value | OR (95% CI) | P-value |
| **28-day mortality** | | | | | | |
| NLR as continuous | 1.50 (1.40–1.60) | <0.001 | 1.36 (1.27–1.45) | <0.001 | 1.16 (1.06–1.27) | <0.001 |
| R1 | Ref | | Ref | | Ref | |
| R2 | 1.85 (1.47–2.35) | <0.001 | 1.61 (1.27–2.05) | <0.001 | 1.34 (1.01–1.80) | 0.048 |
| R3 | 4.76 (3.76–6.08) | <0.001 | 3.34 (2.60–4.31) | <0.001 | 1.95 (1.40–2.75) | <0.001 |
| P for trend | <0.001 | | <0.001 | | <0.001 | |
| **Hospital mortality** | | | | | | |
| NLR as continuous | 1.49 (1.39–1.59) | <0.001 | 1.34 (1.25–1.43) | <0.001 | 1.13 (1.03–1.24) | 0.008 |
| R1 | Ref | | Ref | | Ref | |
| R2 | 1.93 (1.52–2.47) | <0.001 | 1.63 (1.28–2.10) | <0.001 | 1.37 (1.02–1.86) | 0.040 |
| R3 | 4.85 (3.80–6.25) | <0.001 | 3.28 (2.54–4.27) | <0.001 | 1.90 (1.34–2.70) | <0.001 |
| P for trend | <0.001 | | <0.001 | | <0.001 | |
| **ICU mortality** | | | | | | |
| NLR as continuous | 1.44 (1.34–1.54) | <0.001 | 1.30 (1.21–1.39) | <0.001 | 1.14 (1.03–1.25) | 0.008 |
| R1 | Ref | | Ref | | Ref | |
| R2 | 1.79 (1.38–2.33) | <0.001 | 1.48 (1.14–1.95) | 0.004 | 1.31 (0.95–1.82) | 0.109 |
| R3 | 4.25 (3.27–5.58) | <0.001 | 2.82 (2.14–3.75) | <0.001 | 1.79 (1.23–2.62) | 0.003 |
| P for trend | <0.001 | | <0.001 | | <0.001 | |

NLR, Neutrophil-to-Lymphocyte Ratio; R1, low-NLR group; R2, intermediate-NLR group; R3, high-NLR group

Model 1: Unadjusted

Model 2: Adjusted for gender, standardized age, HR, RR, weight, and MAP

Model 3: Adjusted for gender, cerebral infarction, atrial fibrillation, standardized age, HR, RR, weight, MAP, SAPS II, SOFA, 24-hour fluid output, VIS, 24-hour fluid balance, dialysis type, RDW, $PO_2$, HCT, creatinine, lymphocytes, BE, and MCH

## Subgroup analysis

To further evaluate the independent association between the standardized NLR and mortality outcomes, including 28-day, in-hospital, and ICU mortality, we conducted stratified analyses and interaction tests on Model 3 covariates. As demonstrated in Fig 4 and S6 Table, the NLR maintained consistent associations with mortality, with increased predictive value being observed in patients aged >45 years, those with SOFA scores ≤4 and those with SAPS II scores > 29. Those individuals ≥65 years exhibited a 17% increased 28-day mortality risk (P = 0.019), and those with a SOFA score ≤4 exhibited a > 20% increased risk across all of the mortality endpoints (p < 0.001), whereas no significant association was observed in the subgroup of patients with SOFA scores ≥9 (p = 0.369). Significant effect modifications (interaction p < 0.05) were identified in patients with a 24-hour fluid output of 1475–3420 mL, a BE > -2, a HR < 100 bpm, or a RR of 12–20 breaths/min, and similar trends were observed in those with a 24-hour fluid balance ≤2493.27 mL, an MCH ≥ 27 pg, a MAP of 65–110 mmHg, or a creatinine concentration of 1.5–3.0 mg/dL. Although the standardized NLR demonstrated strong associations with all three mortality outcomes in patients with an HR < 60 bpm, the limited sample size within this stratified subgroup resulted in imprecise estimates, thus warranting cautious interpretation. The subgroup analyses consistently demonstrated significant associations between the NLR and all three mortality endpoints across most predefined strata, which confirms the robustness of the predictive value of this biomarker in diverse patient populations.

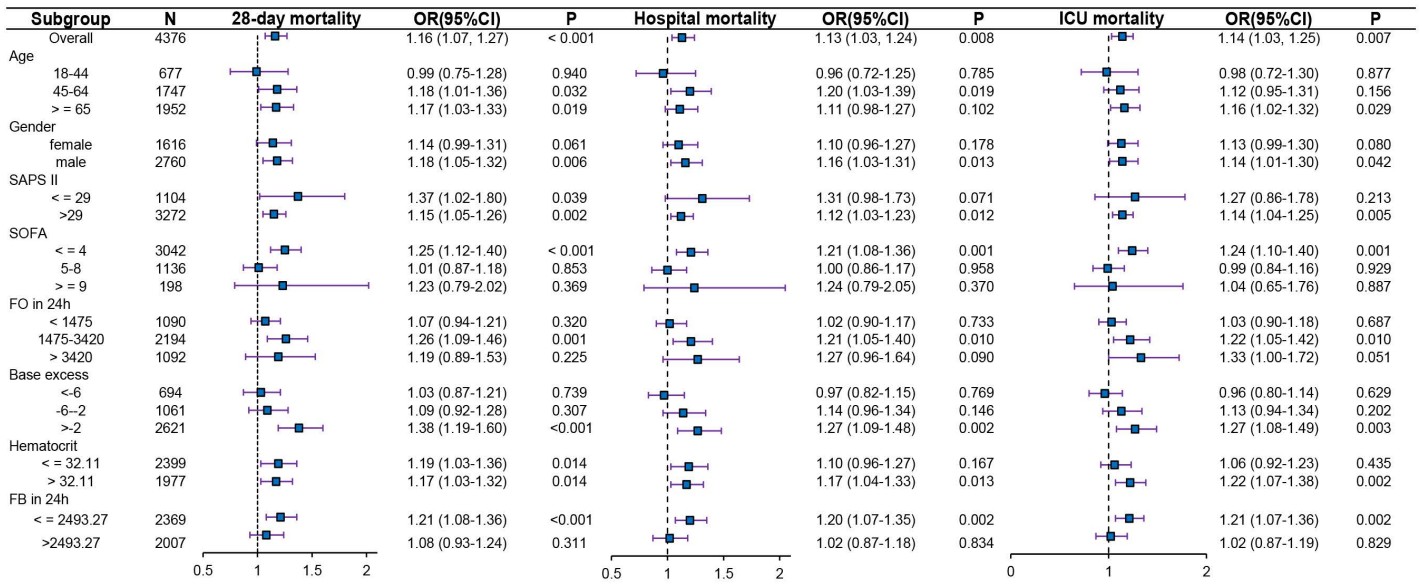

**Fig 4. Forest plot of subgroup analyses for the association between the standardized NLR and mortality outcomes.** The plot displays odds ratios (ORs) with 95% confidence intervals (CIs) for 28-day, in-hospital, and ICU mortality across clinically relevant subgroups. Solid squares represent point estimates of ORs; horizontal lines indicate 95% CIs. The vertical dashed line denotes the null effect (OR = 1).

## Discussion

Machine learning algorithms have increasingly been employed to refine prognostic assessment in sepsis using large-scale ICU databases. For example, Lou et al. [29] and Zheng et al. [30] recently applied XGBoost based on the MIMIC database to explore the association between the triglyceride-glucose (TyG) index and risk of death in sepsis patients, thus further confirming the potential of ML in routine biomarker re-evaluation. Within this evolving landscape, as an accessible and widely recorded inflammatory marker, the NLR has garnered renewed interest; however, its independent prognostic value for short-term mortality remains debated, and systematic ML-driven re-evaluations remain scarce. In the present study, we leveraged a rigorous two-stage feature selection pipeline combining the Boruta algorithm with XGBoost to specifically evaluate the ability of the early NLR to predict 28-day mortality in a large, real-world sepsis cohort. Our optimized model achieved robust discrimination (ROC-AUC 0.875; PR-AUC 0.603), thus outperforming conventional severity scores. More importantly, after full multivariable adjustment, each 1-SD increase in the standardized NLR was independently associated with a 16%, 13%, and 14% higher risk of 28-day, in-hospital, and ICU mortality, respectively. Dose-response analyses confirmed a progressive mortality increase with increasing NLR, whereas RCS revealed a nonlinear, biphasic pattern; specifically, predictive capacity became increased near the first inflection point (NLR = 7.3, approximating the optimal threshold) and plateaued beyond the second inflection point (NLR = 27.3). Subgroup analyses further elucidated that the NLR-mortality association was most pronounced in hemodynamically stable patients and those with low organ failure scores (SOFA ≤ 4), whereas it lost statistical significance in the setting of advanced multi-organ dysfunction (SOFA ≥ 9). Collectively, these findings do not merely replicate prior prognostic observations; rather, they delineate (for the first time) a phenotype-specific, biphasic predictive pattern of the NLR that distinguishes inflammation-driven mortality risk from organ failure-driven mortality. By integrating state-of-the-art feature selection with granular clinical stratification, this study provides a methodologically rigorous and clinically nuanced re-evaluation of the NLR's prognostic utility in sepsis.

Our XGBoost model achieved a test-set ROC-AUC of 0.875 (PR-AUC 0.603) for 28-day mortality prediction, thus representing a substantial improvement over conventional severity scores and individual biomarkers. To contextualize this performance within the broader landscape of emerging prognostic indices in sepsis, we compared our findings with recent studies evaluating the TyG index and the creatinine-to-albumin ratio (CAR) using the MIMIC database [29–31]. Both TyG-based studies reported significant associations with mortality, with Lou et al. [29] identifying a nonlinear relationship (inflection point: TyG = 8.9) and Zheng et al. [30] describing a linear dose-response pattern. The CAR study by Lou et al. [31] demonstrated a J-shaped association with 30-day mortality in sepsis-associated acute kidney injury, with an inflection point at CAR = 1.2 mg/dL and an AUC of 0.68–0.75 being reported. Notably, our NLR-based XGBoost model achieved higher discriminative performance (AUC 0.875) than any of these reported models, which likely reflects the synergistic integration of multiple clinical and laboratory features through a robust two-stage feature selection pipeline. Beyond performance metrics, a key methodological parallel involves the nonlinear risk patterns shared across these biomarkers. The J-shaped CAR-mortality curve and the biphasic NLR-mortality relationship observed in our cohort both suggest threshold effects that demarcate distinct pathophysiological states, including renal dysfunction with systemic inflammation for CAR and innate-adaptive immune imbalance for the NLR. However, the NLR offers distinct practical advantages; specifically, it is derived from a complete blood count without the need for additional metabolic assays, incurs no extra cost, and directly reflects the hyperinflammation–immunosuppression disequilibrium that is a central process to sepsis immunopathology [15,17,18,32]. Thus, although the TyG and CAR capture insulin resistance and renal-inflammatory crosstalk, respectively, the NLR (particularly when embedded in a ML framework) provides complementary prognostic information with superior discrimination and unique biological interpretability. These findings demonstrate the NLR not as a replacement for existing biomarkers; rather, it can function as a phenotype-specific adjunct for early risk stratification in moderate sepsis, where its predictive value is most pronounced.

Recent five-year evidence has consistently demonstrated that an elevated NLR increases sepsis mortality risk by 10–40% across general populations [33–42]. More critically, when high-NLR cohorts were compared with their low-NLR counterparts, multivariable-adjusted analyses revealed that the high-NLR group had an 80% greater mortality risk, which establishes a robust dose–response relationship. Three studies [39,41,42] employing RCS analysis confirmed nonlinear NLR-mortality relationships and demonstrated progressive risk escalation until the NLR reached a range of 15–20, beyond which the trend plateaued. Prior evaluations of the discriminative capacity of the NLR universally relied on ROC curves, with AUROC values ranging from 0.6 to 0.7, which are consistent with our findings and indicate moderate discrimination. However, given the imbalanced outcome distribution (18.4% 28-day mortality in our cohort), ROC metrics may overestimate performance because of excessive true-negative influence, which potentially masks true positive-class identification capacity [32,43,44]. We therefore performed ROC analyses with PR curves and confirmed that the NLR, SOFA score, and SAPS II score all exhibit limited mortality discrimination under outcome imbalance. XGBoost modeling that incorporated multiple parameters significantly improved both the AUC and average precision values. While the NLR demonstrated a substantial overall model contribution, SHAP analysis revealed marginal individual predictive utility, which suggests limited direct mortality risk stratification capacity. Combined NLR stratification and subgroup analyses revealed potential interaction effects, particularly in hemodynamically stable patients for whom the correlations between the NLR and mortality remained robust.

Returning to our primary research question, the limited predictive value of the NLR for 30-day mortality observed in the study by Schupp et al. [19] may stem from two methodological factors. First, the exclusively enrolled critically ill patients demonstrated uniformly high disease severity (mean SOFA score of 11 and mean APACHE II score of 23), and no significant differences were observed in illness severity between survivors and nonsurvivors. Second, the small sample size may have compromised the stability of the results. Conversely, studies [33,35,36,38,39] that support the ability of the NLR to predict sepsis mortality but that demonstrate superior discriminatory performance for 28-day mortality (with the AUROC value being observed at 0.827 in some reports) have predominantly analyzed cohorts with extreme mortality rates

(30–70%) and significant intergroup differences in illness severity scores. Notably, three additional studies reported the limited predictive value of the NLR [37,40,45] after patients with critical illness or hemodynamic instability were excluded through subgroup or sensitivity analyses. These collective observations suggest that the ability of the NLR to predict mortality may be restricted to clinically stable conditions, while its utility appears to decrease in severe cases with concomitant organ dysfunction.

The context-dependent nature of the NLR's prognostic value observed in our study aligns with emerging evidence on other composite biomarkers that integrate inflammatory and nutritional dimensions. Recent studies investigating the NEUT count-to-prognostic nutritional index ratio (NPNR) in septic patients have reported J- or V-shaped associations with mortality [46,47], thus mirroring the biphasic pattern that we identified for the NLR. Both the NLR and NPNR appear to capture the delicate balance between host immune activation and physiological reserve; specifically, their predictive weight diminishes when organ failure becomes the dominant driver of the outcome. This convergence across distinct biomarker constructs reinforces the notion that inflammatory markers are not universal prognostic tools; rather, they are phenotype-specific indicators exhibiting a utility that is most pronounced in moderate disease and wanes in the setting of terminal organ dysfunction. Therefore, future research should focus on context-aware risk stratification that accounts for the dynamic interplay between inflammation, nutrition, and organ function.

Mechanistically, the increase in the NLR in sepsis patients precisely mirrors the dual pathological processes of hyperactivated NEUT and apoptotic LYMPH depletion; thus, the NLR serves as a dynamic biomarker of the "hyperinflammation–immunosuppression" imbalance that characterizes sepsis immunopathology [15,45]. Current evidence indicates that delayed NEUT apoptosis leads to systemic accumulation, whereas accelerated CD4+/CD8+ T-cell apoptosis and impaired NK cell function collectively contribute to this dysregulated state. Notably, the magnitude of NLR elevation may reflect the depletion of anti-inflammatory reserves, and higher values may indicate a poorer capacity to counteract inflammatory insults [45]. Clinically, the NLR demonstrates robust predictive value for moderate sepsis (SOFA scores <9), as evidenced by the finding reported by Liu et al. [37] that serial NLR measurements (AUC = 0.823) outperformed single assessments, as a day-7 NLR > 4.18 predicted significantly increased 28-day mortality. Ye et al. [40] further confirmed that an NLR > 20.25 was an independent predictor of mortality (hazard ratio = 1.22) in large cohorts. However, its ability to predict critical illness (SOFA score ≥11) is diminished, which is likely due to the predominance of multiorgan failure observed in terminal pathophysiology [40]. The primary sources of interstudy variability include discrepancies in measurement timing, divergent threshold selection criteria, and confounding effects of therapeutic interventions such as blood purification. Notably, baseline NLR values inherently differ between specific disease populations and healthy individuals. Karakonstantis et al. [48] reported that multiple factors potentially contribute to nonpathological NLR elevation, including advanced age, exogenous steroid administration, elevated endogenous hormone levels, active hematologic disorders, type 2 diabetes mellitus, and acute trauma. Future research should prioritize three objectives: standardizing measurement protocols, establishing population-specific cutoffs, and investigating combination models with emerging immune biomarkers. Fundamentally, the clinical value of the NLR lies in its noninvasive nature and ability to track immune trajectory evolution rather than as a standalone prognostic determinant.

This study implemented multiple methodological innovations in its investigation of sepsis outcomes. Given the well-documented prognostic influences of key sepsis interventions, including fluid resuscitation, antibiotic therapy, MV, and blood purification [1], we developed advanced data extraction protocols to obtain and systematically categorize relevant parameters from the MIMIC-IV database. Through stepwise feature selection, the analysis consistently identified fluid resuscitation and blood purification as clinically significant factors, while numerous other variables were excluded. A planned evaluation of crystalloid fluid administration effects within the initial three hours after ICU admission was determined to be impractical because of insufficient recorded intravenous infusion volumes. The derived weight-adjusted ratios fell below clinically meaningful thresholds, demonstrated minimal predictive value for mortality outcomes, and were consequently excluded during collinearity analysis and feature selection procedures.

Several limitations must be acknowledged. First, although the MIMIC-IV database provides a large, well-curated sample of critically ill patients, its single-center origin introduces inherent selection bias and reflects the treatment protocols and case mixture of a specific healthcare system. Thus, the absence of external validation represents a substantial limitation. Although we implemented rigorous analytical methods to ensure internal validity (particularly the proposed biphasic, phenotype-dependent predictive pattern of the NLR), our findings must be regarded as hypothesis-generating rather than definitive. Independent validation in prospective, multi-center cohorts is urgently required before the NLR can be considered for routine clinical risk stratification. Furthermore, external validation should specifically test the generalizability of the identified inflection points (NLR = 7.3 and 27.3) and the differential predictive utility across SOFA strata observed in our study. Second, while NLR standardization was necessary to address scale disparities in multivariable regression modeling, this processing may reduce clinical interpretability. Additionally, the exclusive calculation of the NLR from the first available measurement within six hours of ICU admission captures only a static snapshot of the inflammatory state. Given the highly dynamic nature of sepsis, this approach may underestimate the prognostic value of NLR trajectories or peak values, which could better reflect the evolving host response. Future studies incorporating serial NLR measurements or longitudinal trajectory modeling are warranted to further validate our findings. Third, the dataset lacked several potentially relevant parameters, including nutritional status indicators, metabolic markers, and established inflammatory biomarkers such as procalcitonin and C-reactive protein, which may affect the generalizability of the results. Fourth, the sepsis management guidelines were revised multiple times during the extended study period spanning 2008–2022, which may have introduced temporal confounding factors that could have influenced the outcomes. Despite these limitations, this study benefits from a substantial sample size and comprehensive statistical methodology that may partially mitigate these concerns. Future research should focus on targeted analyses of specific sepsis patient subgroups to more precisely characterize the prognostic utility of the NLR.

## Conclusions

This investigation revealed a positive correlation between the magnitude of NLR elevation and sepsis-related mortality, with the strongest predictive value being observed at the highest NLR values. This correlation may primarily reflect the ability of the NLR to predict the outcomes of clinically stable patients, whereas its predictive value may be reduced for patients with severe organ failure. Thus, the utility of the NLR appears to primarily involve dynamic monitoring rather than single static measurements. However, given the absence of external validation, these conclusions remain hypothesis-generating and require confirmation in independent, prospective, multi-center studies before clinical implementation.

## Supporting information

**S1 Fig. Boruta algorithm feature selection results for XGBoost modeling.** Box plots display the Z scores of each parameter, with the x-axis showing their names and the y-axis showing the Z values. Parameters are color-coded by importance (blue for important, green for tentative, and red for unimportant).
(TIF)

**S2 Fig. Feature collinearity assessment in XGBoost modeling.**
(TIF)

**S3 Fig. Unadjusted associations between NLR levels and mortality outcomes.** Panels A-C correspond to 28-day mortality (A), hospital mortality (B), and ICU mortality (C), respectively. Error bars represent 95% CIs, with the intermediate NLR concentration group serving as the reference category.
(TIF)

**S4 Fig. Restricted cubic spline plots illustrating the nonlinear relationship between the NLR and mortality.** (A) 28-day mortality, (B) in-hospital mortality, and (C) ICU mortality. The solid line represents the adjusted odds ratio; the pink band indicates the 95% confidence interval. The histogram displays the distribution of the NLR values, with the median (7.6) and the ROC-derived optimal cutoff (8.08) marked. The nonlinear association was statistically significant for all of the endpoints (P for nonlinearity < 0.05).
(TIFF)

**S1 Table. Missing rate for demographics and clinical variables extracted from the database during the observation period.**
(PDF)

**S2 Table. Comparison of Key Variables Before and After Imputation Using the misForest Algorithm.**
(PDF)

**S3 Table. Selected features and discarded features of collinearity assessment in XGBoost modeling.**
(PDF)

**S4 Table. Performance evaluation of XGBoost model across training and test datasets.**
(PDF)

**S5 Table. Feature importances in the XGBoost model.**
(PDF)

**S6 Table. Subgroup analysis for the association of the NLR with 28-day mortality, hospital mortality and ICU mortality.**
(PDF)

**S7 Table. Abbreviations.**
(PDF)

## Acknowledgments

The authors appreciate all of the investigators who organized, developed, and maintained the MIMIC database.

## Author contributions

**Conceptualization:** Jiyang Liao, Xingwang Chen, Houwang Chen, Huachu Wu, Jianbo Lai.

**Data curation:** Jiyang Liao, Qianwen Xiang, Xingwang Chen.

**Formal analysis:** Jiyang Liao, Xingwang Chen, Long Wu, Houwang Chen, Jianbo Lai.

**Funding acquisition:** Jiyang Liao, Jianbo Lai.

**Investigation:** Jiyang Liao, Qianwen Xiang, Jianbo Lai.

**Methodology:** Jiyang Liao, Xingwang Chen, Huachu Wu, Jianbo Lai.

**Project administration:** Jiyang Liao, Xingwang Chen.

**Resources:** Jiyang Liao, Qianwen Xiang.

**Software:** Jiyang Liao.

**Supervision:** Jiyang Liao, Long Wu, Zhijun Yao, Jianbo Lai.

**Validation:** Jiyang Liao, Zhijun Yao, Huachu Wu.

**Visualization:** Jianbo Lai.

**Writing – original draft:** Qianwen Xiang, Xingwang Chen, Long Wu, Houwang Chen, Huachu Wu, Jianbo Lai.

**Writing – review & editing:** Huachu Wu, Jianbo Lai.

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
