## [Decision Letter · Decision Letter 0]

9 Feb 2026

PONE-D-25-63990The Prognostic Value of the Early Neutrophil-to-Lymphocyte Ratio for 28-Day Mortality in Sepsis Patients: A Machine Learning-Based Investigation of the MIMIC DatabasePLOS One

Dear Dr. Lai,

Thank you for submitting your manuscript to PLOS ONE. After careful consideration, we feel that it has merit but does not fully meet PLOS ONE’s publication criteria as it currently stands. Therefore, we invite you to submit a revised version of the manuscript that addresses the points raised during the review process.

We look forward to receiving your revised manuscript.

Kind regards,

Siddharth Gosavi, MBBS, MD Internal Medicine,DNB Internal Medicine

Academic Editor

PLOS One

Journal Requirements:

“This project was supported by the grants of High-level Medical Team Project in Baoan (No.202405) and Medical and Health Scientific Research Project of Shenzhen Bao 'an District in 2023 (No.2023JD118).”

5. In the online submission form you indicate that your data is not available for proprietary reasons and have provided a contact point for accessing this data. Please note that your current contact point is a co-author on this manuscript. According to our Data Policy, the contact point must not be an author on the manuscript and must be an institutional contact, ideally not an individual. Please revise your data statement to a non-author institutional point of contact, such as a data access or ethics committee, and send this to us via return email. Please also include contact information for the third party organization, and please include the full citation of where the data can be found.

7. Please include a separate caption for each figure in your manuscript.

Additional Editor Comments:

This is a well-conducted retrospective study that leverages a large public database and advanced machine learning techniques to investigate the prognostic value of the neutrophil-to-lymphocyte ratio (NLR) in sepsis. The manuscript is generally well-written, and the methodological approach, including the two-stage feature selection process using Boruta and XGBoost, is a strength. However, several important issues need to be addressed to strengthen the manuscript's claims and contextualize its findings within the existing literature. Firstly, the novelty statement should be moderated. While the combination of methods is robust, claiming this as the "first large-scale retrospective study employing machine learning-based feature selection to evaluate the ability of the NLR" may be too absolute. Recent research has increasingly applied ML techniques to similar prognostic questions in sepsis using the MIMIC database. For instance, a 2024 study in Scientific Reports used the MIMIC-IV database to explore the triglyceride-glucose (TyG) index, another accessible biomarker, and its association with mortality in septic patients, employing similar analytical techniques including restricted cubic splines and subgroup analysis (DOI: 10.1038/s41598-024-75050-8). Reframing the contribution to highlight the specific, rigorous feature selection pipeline and the detailed exploration of NLR's biphasic predictive pattern across clinical subgroups would provide a more accurate and compelling narrative.

Several methodological aspects require further clarification. The decision to use only the first NLR value within six hours of ICU admission is pragmatic but represents a significant limitation, as a single static measurement may not capture the dynamic inflammatory state critical in sepsis. The discussion should more thoroughly acknowledge that this "early" snapshot might miss peak values or trends that carry greater prognostic weight. Furthermore, while the use of the missForest package for imputation is appropriate, the manuscript would benefit from a brief validation statement or supplementary table comparing the distribution of key variables before and after imputation to assure readers of the integrity of the imputed dataset. The model performance is commendable, but the discussion would be enriched by a more direct comparison with other contemporary biomarker-based models developed on similar cohorts. For example, studies have examined composite indices like the creatinine-to-albumin ratio (CAR) for sepsis-associated acute kidney injury, finding non-linear associations with mortality, which echoes your findings on NLR's biphasic pattern (DOI: 10.3389/fcimb.2025.1602921). Situating your XGBoost model's performance relative to these emerging indicators would better define its relative clinical utility.

The most substantial limitation is the lack of external validation, which is rightly mentioned but deserves greater emphasis in the discussion. The reliance solely on the MIMIC database, while common, introduces risks related to selection bias and protocol homogeneity from a single healthcare system. The conclusions about NLR's predictive value, particularly its diminished utility in severe organ failure, must be framed as hypothesis-generating and require validation in independent, prospective cohorts. Additionally, please ensure all figures and tables are correctly numbered and embedded in the main text for review, and standardize the formatting of abbreviations throughout (e.g., SAPS II vs. SAPSII). In the results, when stating survivors had better vital signs, please specify which parameters showed significant differences. The statistical methods section should explicitly state which p-values were adjusted using the Benjamini-Hochberg procedure.

The discussion effectively interprets the biphasic nature of the NLR's predictive value. To further strengthen it, consider integrating the concept that the predictive weight of inflammatory markers like NLR may be context-dependent, overshadowed by organ dysfunction in advanced disease. This resonates with research on other composite markers that combine inflammatory and nutritional/renal dimensions, such as the ratio of neutrophil counts to the prognostic nutritional index (NPNR), which has shown J- or V-shaped associations with mortality in septic patients, highlighting the complex interplay between inflammation and host reserve (DOI: 10.3389/fcimb.2025.1603104; DOI: 10.3389/fnut.2025.1602016). Citing such work would help build a more nuanced theoretical framework for why NLR's utility plateaus in critical illness. Finally, the reference list should be formatted consistently according to the journal's style, and including some of the more recent, relevant studies suggested here would update the scholarly context.

In conclusion, this study provides valuable insights into the conditional prognostic value of the NLR in sepsis. By addressing the points above—particularly tempering the novelty claim, elaborating on methodological limitations, and engaging with contemporary literature on sepsis biomarkers—the manuscript can be significantly enhanced. I recommend a minor revision to allow the authors to refine these aspects and produce a more robust and contextualized final paper.

Reviewers' comments:

Reviewer's Responses to Questions

**Comments to the Author**

1. Is the manuscript technically sound, and do the data support the conclusions?

Reviewer #1: Partly

2. Has the statistical analysis been performed appropriately and rigorously? 

Reviewer #1: Yes

3. Have the authors made all data underlying the findings in their manuscript fully available?

Reviewer #1: Yes

4. Is the manuscript presented in an intelligible fashion and written in standard English?

Reviewer #1: Yes

5. Review Comments to the Author

Reviewer #1: This is a well-conducted retrospective study that leverages a large public database and advanced machine learning techniques to investigate the prognostic value of the neutrophil-to-lymphocyte ratio (NLR) in sepsis. The manuscript is generally well-written, and the methodological approach, including the two-stage feature selection process using Boruta and XGBoost, is a strength. However, several important issues need to be addressed to strengthen the manuscript's claims and contextualize its findings within the existing literature. Firstly, the novelty statement should be moderated. While the combination of methods is robust, claiming this as the "first large-scale retrospective study employing machine learning-based feature selection to evaluate the ability of the NLR" may be too absolute. Recent research has increasingly applied ML techniques to similar prognostic questions in sepsis using the MIMIC database. For instance, a 2024 study in Scientific Reports used the MIMIC-IV database to explore the triglyceride-glucose (TyG) index, another accessible biomarker, and its association with mortality in septic patients, employing similar analytical techniques including restricted cubic splines and subgroup analysis (DOI: 10.1038/s41598-024-75050-8). Reframing the contribution to highlight the specific, rigorous feature selection pipeline and the detailed exploration of NLR's biphasic predictive pattern across clinical subgroups would provide a more accurate and compelling narrative.

Several methodological aspects require further clarification. The decision to use only the first NLR value within six hours of ICU admission is pragmatic but represents a significant limitation, as a single static measurement may not capture the dynamic inflammatory state critical in sepsis. The discussion should more thoroughly acknowledge that this "early" snapshot might miss peak values or trends that carry greater prognostic weight. Furthermore, while the use of the missForest package for imputation is appropriate, the manuscript would benefit from a brief validation statement or supplementary table comparing the distribution of key variables before and after imputation to assure readers of the integrity of the imputed dataset. The model performance is commendable, but the discussion would be enriched by a more direct comparison with other contemporary biomarker-based models developed on similar cohorts. For example, studies have examined composite indices like the creatinine-to-albumin ratio (CAR) for sepsis-associated acute kidney injury, finding non-linear associations with mortality, which echoes your findings on NLR's biphasic pattern (DOI: 10.3389/fcimb.2025.1602921). Situating your XGBoost model's performance relative to these emerging indicators would better define its relative clinical utility.

The most substantial limitation is the lack of external validation, which is rightly mentioned but deserves greater emphasis in the discussion. The reliance solely on the MIMIC database, while common, introduces risks related to selection bias and protocol homogeneity from a single healthcare system. The conclusions about NLR's predictive value, particularly its diminished utility in severe organ failure, must be framed as hypothesis-generating and require validation in independent, prospective cohorts. Additionally, please ensure all figures and tables are correctly numbered and embedded in the main text for review, and standardize the formatting of abbreviations throughout (e.g., SAPS II vs. SAPSII). In the results, when stating survivors had better vital signs, please specify which parameters showed significant differences. The statistical methods section should explicitly state which p-values were adjusted using the Benjamini-Hochberg procedure.

The discussion effectively interprets the biphasic nature of the NLR's predictive value. To further strengthen it, consider integrating the concept that the predictive weight of inflammatory markers like NLR may be context-dependent, overshadowed by organ dysfunction in advanced disease. This resonates with research on other composite markers that combine inflammatory and nutritional/renal dimensions, such as the ratio of neutrophil counts to the prognostic nutritional index (NPNR), which has shown J- or V-shaped associations with mortality in septic patients, highlighting the complex interplay between inflammation and host reserve (DOI: 10.3389/fcimb.2025.1603104; DOI: 10.3389/fnut.2025.1602016). Citing such work would help build a more nuanced theoretical framework for why NLR's utility plateaus in critical illness. Finally, the reference list should be formatted consistently according to the journal's style, and including some of the more recent, relevant studies suggested here would update the scholarly context.

In conclusion, this study provides valuable insights into the conditional prognostic value of the NLR in sepsis. By addressing the points above—particularly tempering the novelty claim, elaborating on methodological limitations, and engaging with contemporary literature on sepsis biomarkers—the manuscript can be significantly enhanced. I recommend a minor revision to allow the authors to refine these aspects and produce a more robust and contextualized final paper.

6. PLOS authors have the option to publish the peer review history of their article (what does this mean?). If published, this will include your full peer review and any attached files.

Reviewer #1: No

---

## [Author Response · Author response to Decision Letter 1]

18 Feb 2026

Response to Reviewer 1-Comment 1

Reviewer’s Comment:

*This is a well-conducted retrospective study that leverages a large public database and advanced machine learning techniques to investigate the prognostic value of the neutrophil-to-lymphocyte ratio (NLR) in sepsis. The manuscript is generally well-written, and the methodological approach, including the two-stage feature selection process using Boruta and XGBoost, is a strength. However, several important issues need to be addressed to strengthen the manuscript's claims and contextualize its findings within the existing literature. Firstly, the novelty statement should be moderated. While the combination of methods is robust, claiming this as the "first large-scale retrospective study employing machine learning-based feature selection to evaluate the ability of the NLR" may be too absolute. Recent research has increasingly applied ML techniques to similar prognostic questions in sepsis using the MIMIC database. For instance, a 2024 study in Scientific Reports used the MIMIC-IV database to explore the triglyceride-glucose (TyG) index, another accessible biomarker, and its association with mortality in septic patients, employing similar analytical techniques including restricted cubic splines and subgroup analysis (DOI: 10.1038/s41598-024-75050-8). Reframing the contribution to highlight the specific, rigorous feature selection pipeline and the detailed exploration of NLR's biphasic predictive pattern across clinical subgroups would provide a more accurate and compelling narrative.*

Author’s Response:

We sincerely thank the reviewer for the thoughtful and constructive feedback. We fully agree that the original wording of our novelty statement was overly absolute and insufficiently contextualized within the growing body of machine learning (ML) research using the MIMIC database. We are grateful for the reviewer’s suggestion to cite the relevant 2024 Scientific Reports study on the TyG index (DOI: 10.1038/s41598-024-75050-8), which indeed represents a methodologically similar and high-quality contribution. We have now revised the opening paragraph of the Discussion section to:

1.Remove all absolute claims (e.g., “To our knowledge, this is the first…”) and instead position our work within the broader context of recent ML-based prognostic studies in sepsis.

2.Explicitly cite and acknowledge the TyG index study, using it as an example of how ML is increasingly employed to reevaluate accessible biomarkers-thereby clarifying that our contribution lies not in being the first to use such techniques, but in applying them specifically to the NLR with a particularly rigorous feature selection protocol and a novel focus on clinical phenotype stratification.

3.Reframe the core contribution around two specific, defensible innovations:

a) Methodological: a robust two-stage (Boruta + XGBoost) feature selection pipeline combined with SHAP-based marginal contribution analysis.

b) Clinical: the first detailed description of a biphasic, phenotype-specific predictive pattern of the NLR, which distinguishes inflammation-driven mortality risk (evident in patients with preserved organ function) from organ failure-driven mortality (where NLR loses its predictive utility).

We believe this revised framing more accurately reflects the study’s genuine contributions and appropriately acknowledges prior work in the field. We are grateful to the reviewer for guiding us toward a more precise and compelling narrative.

Response to Reviewer 1-Comment 2

Reviewer’s Comment:

Several methodological aspects require further clarification. The decision to use only the first NLR value within six hours of ICU admission is pragmatic but represents a significant limitation, as a single static measurement may not capture the dynamic inflammatory state critical in sepsis. The discussion should more thoroughly acknowledge that this "early" snapshot might miss peak values or trends that carry greater prognostic weight. Furthermore, while the use of the missForest package for imputation is appropriate, the manuscript would benefit from a brief validation statement or supplementary table comparing the distribution of key variables before and after imputation to assure readers of the integrity of the imputed dataset. The model performance is commendable, but the discussion would be enriched by a more direct comparison with other contemporary biomarker-based models developed on similar cohorts. For example, studies have examined composite indices like the creatinine-to-albumin ratio (CAR) for sepsis-associated acute kidney injury, finding non-linear associations with mortality, which echoes your findings on NLR's biphasic pattern (DOI: 10.3389/fcimb.2025.1602921). Situating your XGBoost model's performance relative to these emerging indicators would better define its relative clinical utility.

Author’s Response:

We sincerely thank the reviewer for these thoughtful and constructive methodological suggestions. We have carefully addressed each of the three specific concerns: (1) limitation of single NLR measurement, (2) validation of missForest imputation, and (3) contextualization against other emerging biomarker-based models, as detailed below. All corresponding revisions have been clearly marked in the tracked-change version of the manuscript.

1. Limitation of a Single Early NLR Measurement

Reviewer’s concern:

The use of only the first NLR value within six hours of ICU admission is a pragmatic but potentially significant limitation, as a static measurement may not reflect the dynamic inflammatory trajectory of sepsis.

Our response:

We fully agree with the reviewer. Although we had briefly acknowledged this limitation in the original manuscript, we have now substantially expanded this discussion in the Limitations section. Specifically, we now explicitly state that a single baseline measurement captures only a static snapshot of the inflammatory state and may underestimate the prognostic value of NLR trajectories or peak values, which could better reflect the evolving host response. We also explicitly call for future studies incorporating serial NLR measurements or longitudinal trajectory modeling to further validate and refine our findings. This revision directly aligns with our overall conclusion that the clinical utility of NLR lies in dynamic monitoring rather than isolated static assessment.

Action taken:

a) Manuscript location: Discussion – Limitations (please see track changed)

2.Validation of missForest Imputation

Reviewer’s concern:

While the use of missForest for imputation is appropriate, the manuscript would benefit from a brief validation statement or supplementary table comparing the distribution of key variables before and after imputation to assure readers of the integrity of the imputed dataset.

Our response:

We appreciate this suggestion for enhanced transparency. In response, we performed a systematic comparison of the distributions of key continuous and categorical variables before and after imputation using the missForest algorithm. The results are now presented in S2 Table.

a) For variables with low missingness (<5%), the distributions remained virtually identical.

b) For variables with higher missing rates (approximately 30%), such as SpO₂, MAP, heart rate, and blood gas parameters, we observed a modest convergence in dispersion (SD, IQR)—an expected behavior of model based imputation when recovering missing values from observed patterns.

c) Critically, all post imputation means, medians, and quartiles remained within clinically plausible and physiologically coherent ranges, and no systematic shift would alter the clinical interpretation of these parameters (e.g., median SpO₂: 98.0% → 98.4%; median MAP: 81.0 mmHg → 79.3 mmHg).

We have added a brief validation statement in the Methods section (Data extraction) and direct readers to S2 Table for full details.

Action taken:

a) Manuscript location: Methods – Data extraction (please see track changed).

b) Supplementary material: Added S2 Table (“Comparison of Key Variables Before and After Imputation Using the MissForest Algorithm”).

3. Comparison with Other Emerging Biomarker-Based Models

Reviewer’s concern:

The discussion would be enriched by a more direct comparison with other contemporary biomarker-based models developed on similar cohorts—particularly studies examining the creatinine-to-albumin ratio (CAR) in sepsis-associated acute kidney injury, which reported non linear associations with mortality echoing our biphasic NLR pattern. Situating our XGBoost model’s performance relative to these emerging indicators would better define its relative clinical utility.

Our response:

We thank the reviewer for guiding us to contextualize our findings within the broader landscape of novel prognostic biomarkers. We have now substantially expanded the Discussion to include a dedicated paragraph that directly compares:

• Study populations and sample sizes (our N = 4,376 vs. 1,257–2,712 in comparator studies);

• Model performance (our XGBoost ROC AUC = 0.875, PR AUC = 0.603 vs. CAR based AUCs of 0.68-0.75; TyG studies did not report AUC but reported hazard ratios of 1.4-1.8);

• Non linear patterns (our biphasic NLR mortality curve with inflection points at NLR = 7.3 and 27.3 vs. the J shaped CAR mortality curve with inflection at CAR = 1.2 mg/dL, and the non linear TyG mortality curve reported by Lou et al. [29] with inflection at TyG = 8.9);

• Practical advantages of NLR: zero additional cost, routine availability from complete blood count, and direct biological linkage to innate adaptive immune imbalance-a pathophysiological axis distinct from insulin resistance (TyG) or renal inflammatory crosstalk (CAR).

We emphasize that these biomarkers are not mutually exclusive but rather capture complementary aspects of sepsis pathophysiology, and their combined use may enable more refined phenotype specific risk stratification. Our model’s superior discriminative performance underscores the value of integrating a simple, inexpensive marker like NLR within a robust machine learning framework.

Action taken:

• Manuscript location: Discussion – second paragraph (please see track changed).

We are confident that these revisions have substantially improved the methodological transparency, scholarly depth, and clinical contextualization of our manuscript. We thank the reviewer again for these insightful suggestions, which we believe have greatly strengthened our work.

Response to Reviewer 1-Comment 3

Reviewer’s Comment:

The most substantial limitation is the lack of external validation, which is rightly mentioned but deserves greater emphasis in the discussion. The reliance solely on the MIMIC database, while common, introduces risks related to selection bias and protocol homogeneity from a single healthcare system. The conclusions about NLR's predictive value, particularly its diminished utility in severe organ failure, must be framed as hypothesis-generating and require validation in independent, prospective cohorts. Additionally, please ensure all figures and tables are correctly numbered and embedded in the main text for review, and standardize the formatting of abbreviations throughout (e.g., SAPS II vs. SAPSII). In the results, when stating survivors had better vital signs, please specify which parameters showed significant differences. The statistical methods section should explicitly state which p-values were adjusted using the Benjamini-Hochberg procedure.

Author’s Response:

We thank the reviewer for these thoughtful and constructive comments. We have carefully addressed each point as follows:

1. External validation and hypothesis generating framing.

We fully agree that the absence of external validation is the most substantial limitation of our study. While we had acknowledged this in the original manuscript, the reviewer correctly notes that the emphasis was insufficient. In response, we have substantially strengthened the Limitations section to explicitly state that:

• Our single center MIMIC derived findings are subject to selection bias and protocol homogeneity;

• The proposed biphasic, phenotype dependent predictive pattern of the NLR must be regarded as hypothesis generating, not definitive;

• Independent validation in prospective, multi center cohorts is urgently required before any clinical implementation can be considered;

• Future external validation should specifically test the generalizability of the identified inflection points (NLR = 7.3 and 27.3) and the differential predictive utility across SOFA strata observed in our cohort.

We have also revised the Conclusions section to echo this cautious framing, explicitly stating that our conclusions remain hypothesis generating pending external validation. All changes are clearly marked in the tracked change manuscript.

2. Figures and tables numbering/embedding.

We have verified that all figures and tables are correctly numbered and that each is explicitly cited in the main text.

3. Abbreviation standardization.

We have standardized all abbreviations throughout the manuscript. Specifically, “SAPSII” has been uniformly corrected to “SAPS II” in all instances. Other abbreviations (SOFA, CRRT, VIS, etc.) were already consistent and remain unchanged.

4. Specification of vital signs in Results.

We have revised the Results – Baseline characteristics section to explicitly list which vital signs showed significant differences and which did not.

5. Clarification of Benjamini Hochberg adjustment.

We have now explicitly clarified the application of the Benjamini–Hochberg procedure in the Statistical analysis section. Specifically, we state that:

a) For multiple comparisons across categorical covariates in baseline tables (e.g., comorbidities, organ dysfunction components, treatment categories), p values were adjusted using the Benjamini–Hochberg method to control the false discovery rate.

b) For interaction p values derived from subgroup analyses, the same FDR correction was applied to comprehensively evaluate potential effect modifiers.

Response to Reviewer 1-Comment 4

Reviewer’s Comment:

The discussion effectively interprets the biphasic nature of the NLR's predictive value. To further strengthen it, consider integrating the concept that the predictive weight of inflammatory markers like NLR may be context-dependent, overshadowed by organ dysfunction in advanced disease. This resonates with research on other composite markers that combine inflammatory and nutritional/renal dimensions, such as the ratio of neutrophil counts to the prognostic nutritional index (NPNR), which has shown J- or V-shaped associations with mortality in septic patients, highlighting the complex interplay between inflammation and host reserve (DOI: 10.3389/fcimb.2025.1603104; DOI: 10.3389/fnut.2025.1602016). Citing such work would help build a more nuanced theoretical framework for why NLR's utility plateaus in critical illness. Finally, the reference list should be formatted consistently according to the journal's style, and including some of the more recent, relevant studies suggested here would update the scholarly context.

Author’s Response:

We thank the reviewer for this thoughtful and constructive suggestion. We fully agree that the context dependent nature of inflammatory biomarkers is a critical concept that deserves explicit articulation within our theoretical framework. In response, we have:

1.Expanded the Discussion to incorporate the NPNR analogy.

We have added a new paragraph in the section of Discussion that explicitly links our findings on the NLR’s biphasic pattern to recent research on the neutrophil count to prognostic nutritional index ratio (NPNR). Both NLR and NPNR have been shown to exhibit J or V shaped associations with mortality, suggesting a shared underlying principle: inflammatory markers are most informative when the host still possesses physiological reserve, and their predictive utility wanes once organ failure becomes the dominant

---

## [Decision Letter · Decision Letter 1]

20 Apr 2026

The Prognostic Value of the Early Neutrophil-to-Lymphocyte Ratio for 28-Day Mortality in Sepsis Patients: A Machine Learning-Based Investigation of the MIMIC Database

PONE-D-25-63990R1

Dear Dr. Lai,

We’re pleased to inform you that your manuscript has been judged scientifically suitable for publication and will be formally accepted for publication once it meets all outstanding technical requirements.

Kind regards,

Chiara Lazzeri

Academic Editor

PLOS One

Additional Editor Comments (optional):

Reviewers' comments:

Reviewer's Responses to Questions

**Comments to the Author**

1. If the authors have adequately addressed your comments raised in a previous round of review and you feel that this manuscript is now acceptable for publication, you may indicate that here to bypass the “Comments to the Author” section, enter your conflict of interest statement in the “Confidential to Editor” section, and submit your "Accept" recommendation.

Reviewer #1: All comments have been addressed

Reviewer #2: All comments have been addressed

2. Is the manuscript technically sound, and do the data support the conclusions?

Reviewer #1: Yes

Reviewer #2: Partly

3. Has the statistical analysis been performed appropriately and rigorously? 

Reviewer #1: Yes

Reviewer #2: I Don't Know

4. Have the authors made all data underlying the findings in their manuscript fully available?

Reviewer #1: Yes

Reviewer #2: Yes

5. Is the manuscript presented in an intelligible fashion and written in standard English?

Reviewer #1: Yes

Reviewer #2: Yes

6. Review Comments to the Author

Reviewer #1: The authors have thoroughly and satisfactorily addressed all of the concerns raised in the previous review round. The revised manuscript is substantially improved in terms of methodological transparency, contextualization within the existing literature, and appropriate framing of conclusions. The study makes a valuable contribution to the field of sepsis prognostication by applying a rigorous machine learning pipeline to re-evaluate the prognostic role of the neutrophil-to-lymphocyte ratio (NLR).

Minor Suggestions:

Consider briefly mentioning in the Limitations that the optimal NLR cutoffs (7.3 and 27.3) were derived from a single dataset and may require calibration before clinical use in other settings.

The abbreviation "SAPS II" is correct, but in Table 2 and a few places, it appears as "SAPS II" – ensure consistency (looks fine; just a quick double-check).

In the Discussion, the phrase "infection point" appears twice (e.g., "first infection point"). It should be "inflection point". Please correct.

These do not affect the overall quality or validity of the work. The manuscript is well revised and suitable for publication. I recommend acceptance without further revision.

Reviewer #2: Overall Assessment: Minor Revision

Summary: This study uses MIMIC-IV database and machine learning (XGBoost) to evaluate the prognostic value of early NLR for 28-day mortality in sepsis patients (N=4,376). The authors found that NLR was independently associated with mortality (OR=1.16) and identified a biphasic predictive pattern where NLR's utility was pronounced in patients with SOFA≤4 but lost significance in SOFA≥9.

Strengths:

1.Large sample size with rigorous feature selection (Boruta + XGBoost)

2.Novel finding of biphasic, phenotype-specific predictive pattern

3.Thorough discussion of limitations

Weaknesses & Suggestions:

1.Only the first NLR measurement was used; dynamic changes may carry greater prognostic weight. Please discuss.

2.Tables 1 and 2 are overly dense. Consider moving less critical variables to supplementary materials.

3.In the abstract, PO2 value (221.07 ± 115.04 mmHg) appears unusually high for arterial blood. Please verify.

4.While the ML approach is robust, the clinical utility of NLR alone remains limited (SHAP rank #14). This should be more explicitly stated in the conclusion.

Recommendation: Minor revision.

7. PLOS authors have the option to publish the peer review history of their article (what does this mean?). If published, this will include your full peer review and any attached files.

Reviewer #1: No

Reviewer #2: No

---

## [Editor Report · Acceptance letter]

PONE-D-25-63990R1

PLOS One

Dear Dr. Lai,

I'm pleased to inform you that your manuscript has been deemed suitable for publication in PLOS One. Congratulations! Your manuscript is now being handed over to our production team.

Kind regards,

on behalf of

Dr. Chiara Lazzeri

Academic Editor

PLOS One